# Evaluation of RNA Stability and Molecular Biomarkers for Post-Mortem Interval Estimation in Rat Organs

**DOI:** 10.3390/ijms262211227

**Published:** 2025-11-20

**Authors:** Minju Jung, Sujin Choi, Mingyoung Jeong, Sohyeong Kim, Dong Geon Lee, Kwangmin Park, Xianglan Xuan, Yujin Park, Heechul Park, Dong Hyeok Kim, Jungho Kim, Min Ho Lee, Yoonjung Cho, Sunghyun Kim

**Affiliations:** 1Department of Clinical Laboratory Science, College of Health Sciences, Catholic University of Pusan, Busan 46252, Republic of Korea; jungm9911@naver.com (M.J.); hyeongso29@naver.com (S.K.); ehdrjs6789@naver.com (D.G.L.); pkmchi777@naver.com (K.P.); xuan929@naver.com (X.X.); qkr8051@naver.com (Y.P.); jutosa70@cup.ac.kr (J.K.); 2Next-Generation Industrial Field-Based Specialist Program for Molecular Diagnostics, Brain Busan 21 Plus Program, Graduate School, Catholic University of Pusan, Busan 46252, Republic of Korea; 3Department of Forensic Science, Graduate School, Catholic University of Pusan, Busan 46252, Republic of Korea; tnwls7859-@naver.com (S.C.); 5585mk@naver.com (M.J.); 4Department of Clinical Laboratory Science, Hyejeon University, Hongseong 32244, Republic of Korea; phc2626@hj.ac.kr; 5Department of Biomedical Laboratory Science, Masan University, Changwon 51217, Republic of Korea; ehd6091@naver.com; 6Forensic DNA Division, Gwangju Institute, National Forensic Service, Jangseong 57248, Republic of Korea; lmh77777@naver.com

**Keywords:** post-mortem interval (PMI), histological changes, RNA biomarkers, reverse transcription quantitative PCR, organs

## Abstract

Post-mortem interval (PMI) analysis plays a crucial role in forensic investigations, providing essential insights into the time since death. This study examined histological changes and ribonucleic acid (RNA) quantification across major organs to identify molecular indicators for PMI estimation. Because RNA gradually degrades after death, understanding its stability under different tissue, temperature, and PMI conditions provides valuable forensic insights. We analyzed post-mortem changes in total RNA from the heart, kidney, liver, and lung tissues of Sprague Dawley rats stored at 4 °C and 26 °C. Tissue samples were collected at various PMIs and evaluated histologically for cellular integrity. Total RNA concentration and purity were measured, and complementary DNA (cDNA) was synthesized for molecular analysis. Expression levels of 5S rRNA, *B2m*, *Gapdh*, and *Sort1* were quantified using reverse transcription quantitative PCR (RT-qPCR). The results showed that PMI and organ type significantly affected total RNA concentration, whereas temperature exerted only a minor effect. Among the four target genes, 5S rRNA exhibited the lowest Ct values, indicating the highest stability. Notably, RNA degradation patterns varied with temperature, particularly in kidney and liver tissues. These findings suggest that RNA-based molecular markers, particularly 5S rRNA, may serve as promising indicators for accurate PMI estimation.

## 1. Introduction

The post-mortem interval (PMI) is a crucial parameter in forensic investigations, serving as a fundamental tool for determining the time of death, identifying unknown bodies, and clarifying causes of death. It is particularly important in cases of suicide, accident, and unexplained death [1]. Therefore, accurate estimation of PMI remains a major focus of forensic research, with continuous technological progress improving precision and reliability.

Recent advances in forensic science have introduced innovative approaches that enhance PMI estimation accuracy [2]. Traditional methods—such as assessments of *rigor* mortis, *livor* mortis, entomology analysis, Henssge’s nomogram, microbial ecology, imaging and ultrasound, and evaluation of the corpse’s physical condition—remain valuable tools [3]. Forensic entomology can identify the PMI as a variation in maggots’ or insect species’ life cycles, ecological presence in each environment, and clusters [4]. Henssge’s nomogram estimates the PMI through the rectal temperature of corpses. Microbial ecology is utilized to estimate the PMI by studying microbial communities around the corpse [5]. Changes in microbial communities are closely related to the decomposition process and used to predict accurate PMIs [6]. The condition of the corpse, including *pallor* mortis, *algor* mortis, *rigor* mortis, *livor* mortis, putrefaction, decomposition, skeletonization, and fossilization, progresses sequentially; therefore, the PMI can be measured in sequence [7]. However, each factor might be different depending on the environment surrounding where the corpse is placed; therefore, it is difficult to derive an accurate PMI. The ambient climate, such as the temperature and humidity, influence PMI estimation. Lower temperatures slow down decomposition, whereas higher temperatures could accelerate it. Therefore, temperature information is a critical factor in PMI estimation models [8]. Humidity also affects the PMI. In high-humidity environments, there might be an acceleration of decomposition due to increased microbial activity. Microorganisms thrive in moist conditions, leading to faster degradation of body tissues [9]. On the other hand, in low-humidity environments, the decomposition process could slow down, because moisture is limited, affecting the activities of the decomposing organisms. Therefore, when estimating or determining the PMI, it is crucial to consider the environment’s conditions, including humidity, since they could play a significant role in the post-mortem changes that occur. Researchers and forensic experts often consider a combination of environmental factors, including temperature and humidity, to improve the accuracy of PMI estimations [10].

With an increase in PMI, histological changes occur in body tissues [11]. As the processes of decomposition and decay progress, various histological changes can manifest, and these may vary depending on the PMI. Some common histological changes include cellular disintegration, nuclear condensation (pyknosis), karyorrhexis, loss of membrane integrity, tissue liquefaction, and microbial infiltration [12]. Cell breakdown and disintegration occur when the stable structure of cells becomes unstable, leading to disintegration of the cells. Alterations to cell nuclei may become pyknotic and undergo morphological changes. Cell proliferation and tumor formation take place when rapid cell proliferation and the formation of tumors can occur due to decomposition and decay. Blood coagulation and vascular rupture can lead to changes in tissue color. Changes due to microbial activity occur when the activity of bacteria and fungi lead to the structural instability and deformation of tissues. The decomposition of fat cells and connective tissues lead to the tissues’ liquefaction and increased instability. These histological changes associated with the PMI could vary depending on the process of decay and the environmental conditions surrounding the body. Therefore, in forensic investigations, it is crucial to assess and interpret histological changes in consideration of the time elapsed since death [13].

The present study focuses on total RNA extraction, which is widely used in forensic genetics, from major organ tissues. Forensic genetics could be analyzed from various sources such as blood, saliva, teeth, bones, organs, and semen, etc. Therefore, the main advantage is that DNA and RNA can be extracted regardless of the condition of the corpse. It can be seen that the quality and concentration of total RNA are obtained according to the PMI, as well as the housekeeping gene and 5S rRNA expression level, are detected. The cycle of threshold (Ct) values from each biomarker generally increased with prolonged PMI. The quality and concentration of total RNA and the expression levels of housekeeping genes were observed and 5S rRNA, according to the PMI, demonstrated that RNA degrades faster than DNA in dead cells [14]. In other words, RNA remains stable for a shorter time than immediately after death. Therefore, RNA could be used as useful biomarker for short-term PMI estimation to rapidly detect the activation or suppression of specific genes associated with changes in a decomposing body. There were experiments conducted with SD rats using 5S rRNA, an endogenous reference gene, as well as *B2m*, *Gapdh*, and *Sort1*, which are housekeeping genes of SD rats [15,16]. According to these studies, this enables quick tracking of body changes and aids in PMI estimation. As a result, RNA is utilized as a valuable tool in some PMI estimation methods to provide faster results and more accurate tracking of body changes compared to DNA.

Recently, small non-coding RNAs such as microRNAs (miRNAs) and circular RNAs (circRNAs) have emerged as promising molecular markers for PMI estimation. Their relative stability in degraded tissues allows for extended windows compared with mRNA [17]. Integrating these RNA species into PMI models may improve accuracy and expand applicability to different environmental conditions. However, many previous studies were limited by narrow PMI ranges or constant temperature conditions or focused primarily on DNA-based approaches [18].

Therefore, the present study was designed to comprehensively evaluate post-mortem molecular and histological changes under two temperature conditions (4 °C and 26 °C) and multiple PMI time points using rat heart, kidney, liver, and lung tissues. Total RNA purity and quantity, as well as the expression of selected housekeeping genes and rRNA, were analyzed by histology, RNA quantification, and reverse transcription quantitative PCR (RT-qPCR) SYBR Green assay to explore their correlation with the PMI.

## 2. Results

### 2.1. Changes in the Tissue Morphology According to PMI Under Different Temperature Conditions

The samples for microscopic analysis were prepared at different PMIs (0 h, 24 h, 48 h, 4 days, 8 days, and 21 days). Heart, kidney, liver, and lung tissues were sectioned and H&E stained, and they showed a progressive loss of nuclear material over time. This progressive loss of nuclei was represented in all the replicates analyzed.

After 8 days, most tissue samples at 26 °C began to exhibit signs of degradation, with the dye intensity reduced in line with the level of nuclear material. The overall structure was beginning to break down with large open spaces forming. The 21-day samples exhibited complete degradation with no recognizable tissue features. At 4 °C, the nucleus remained intact until 21 days for all tissues (Figure 1, Figure 2, Figure 3 and Figure 4).

### 2.2. Changes in Ct Value from RT-qPCR SYBR Green Assay According to PMI Under Temperature Conditions

For more detail, the Ct values were analyzed using a RT-qPCR SYBR Green assay and the results are as follows. The comparison of the Ct value of the 5S rRNA results across the heart, kidney, liver, and lung groups revealed significant differences, as shown in Figure 5. First, Figure 5 shows there was a difference in the Ct values of the liver samples at different temperatures according to the PMI. In the case of the liver at 26 °C, the Ct value started to decrease from 48 h while it remained stable at 4 °C. In the kidney, the difference between the Ct values at 26 °C and 4 °C could be seen over time. In particular, a clear difference could be seen after 8 days.

The comparison of the Ct values of the *B2m* and Gapdh results for the four tissue groups revealed no significant differences, as shown in Figure 6 and Figure 7. Figure 6 and Figure 7 shows there was no significant change in the four collected tissue samples at 26 °C and 4 °C.

The comparison of the Ct values of the *Sort1* results for the four tissues revealed no significant difference, as shown in Figure 8. Figure 8 shows that the Ct values remained constant in the liver samples regardless of PMI and temperature.

Overall, 5S rRNA is ribosome-related RNA, which combines specific proteins to form a complex, so it has high stability and can be confirmed that the Ct value is lower than other housekeeping genes. *B2m* and Sort1 can see that the Ct value remained constant, and 5S rRNA and *Gapdh* can see that the Ct value varies with temperature, long-term, and post-mortem time.

### 2.3. Changes in Delta Ct Values from RT-qPCR SYBR Green Assay According to PMI Under Different Temperature Conditions

As observed in the previous results, 5S rRNA was stable at different PMI time points. Therefore, 5S rRNA was used as an endogenous control gene to adjust the mRNA expression of the three target genes. The delta Ct value, obtained by subtracting the Ct value of the endogenous control gene from the target gene, was created and plotted. The delta Ct value is used to estimate changes in gene expression or to compare differences in the expression levels of multiple samples. The delta Ct value was calculated to normalize the expression level of each target gene to that of the endogenous control gene. It was defined as delta Ct = Ct(target) − Ct(reference), where Ct(reference) represents the Ct value of 5S rRNA. This calculation allows for a comparison of relative expression levels among different PMIs and tissues. A positive delta Ct value indicates lower expression of the target gene relative to the reference gene, whereas a smaller or negative delta Ct value indicates higher expression. The delta Ct method is widely used to assess changes in gene expression under varying experimental conditions. This value is mainly utilized for a comparison of relative expression levels and helps us to understand how the expression of a particular gene changes under a given experimental condition. A positive delta Ct value indicates a target at different PMI time points. Therefore, 5S rRNA was used as an endogenous control gene to adjust the mRNA expression of the three target genes. The delta Ct value, obtained by subtracting the Ct value of the endogenous control gene from the target gene, was created and plotted. The delta Ct value is used to estimate changes in gene expression, or to compare differences in the expression levels of multiple samples. This value is mainly utilized for a comparison of relative expression levels and helps us understand how the expression of a particular gene changes under a given experimental condition. A positive delta Ct value indicates that the target gene is expressed at a relatively high level compared to the reference gene, while a negative value indicates that it is expressed at a low level.

Each graph shows the result divided by temperature and tissues. In all results, the delta Ct values of *Gapdh* show the lowest value, and those of *B2m* and *Sort1* show similar patterns (Figure 9, Figure 10, Figure 11 and Figure 12). In the 26 °C lungs, the delta Ct value decreased in *Gapdh* as the PMI progressed (Figure 12).

## 3. Discussion

The PMI is a critical parameter in forensic investigations and has been extensively studied [19]. Moreover, research on forensic genomics for its implications in PMI is being continued [20]. In particular, many studies on the effects of temperature and environmental conditions are being actively conducted [21].

Recent research aims to establish accurate PMI estimation methods, particularly through forensic genomics [22]. However, human studies lack standardized post-mortem conditions, while rat experiments are limited by short PMIs and fixed temperatures.

The present study compared the pattern of molecular biomarkers according to the PMI by extracting total RNA from rat tissues, which degrade faster than DNA. In particular, we applied a wide range of PMI conditions from 6 h to 21 days, as well as temperature conditions of 26 °C and 4 °C, to observe histological structures and markers such as *B2m*, *Gapdh*, and *Sort1*.

The results revealed that most of the four organs containing heart, kidney, liver, and lung showed no nucleus or a lot of collapse after 8 days at 26 °C, but at 4 °C, the nucleus remained in shape until 21 days. Based on these histological results, total RNA was extracted for a comparison with the molecular results, including the Ct and delta Ct values. 5S rRNA is a structural RNA component with low Ct values, potentially resisting degradation by intracellular exonucleases. Ct values of 5S rRNA showed differences between 26 °C and 4 °C after 48 h in the kidney and the liver. Unexpectedly, Ct values for 5S rRNA decreased in the liver at 26 °C after 48 h, suggesting apparent RNA increase. This may reflect selective preservation or technical artifacts rather than genuine stabilization. Further experiments are needed to clarify whether this is due to reduced RT-qPCR inhibition or differences in RNA extraction efficiency over time. The genes selected in this study—5S rRNA, *B2m*, *Gapdh*, and *Sort1*—were chosen because they represent different molecular characteristics that may influence their post-mortem stability. 5S rRNA is a structural RNA component, relatively resistant to degradation, whereas *B2m* and *Gapdh* are commonly used housekeeping genes with stable expression under various physiological conditions. Sort1, a membrane-associated gene, was included to explore the behavior of a less conventional reference gene under post-mortem conditions. Differences in RNA degradation among these markers may reflect variations in transcript structure, localization, and protection by ribonucleoprotein complexes. 5S rRNA was chosen as a reference gene for its structural stability; however, as it is transcribed by RNA polymerase III, it may not be ideal for normalizing mRNA targets transcribed by RNA polymerase II. This difference may limit its suitability as a normalization reference for mRNA quantification. Moreover, the assumption that 5S rRNA remains constant across different PMIs and temperature conditions was not experimentally verified in this study. Therefore, future studies should experimentally validate the stability of 5S rRNA using commonly employed algorithms such as geNorm, NormFinder, or BestKeeper. Alternatively, incorporating multiple reference genes with proven stability under post-mortem conditions would strengthen normalization accuracy. This will help to confirm whether 5S rRNA is appropriate for PMI-related analyses or if mRNA-derived housekeeping genes provide a more reliable reference framework.

Delta Ct values were calculated by subtracting the reference gene from the target gene. In the 26 °C heart, *Gapdh* delta Ct values showed a gradual decrease according to PMI. In the 4 °C heart, there was a decrease in delta Ct values after 8 days. Overall, under 4 °C conditions, *B2m*, *Gapdh*, and *Sort1* remained relatively constant without significant changes. In the 26 °C lung, *Gapdh* delta Ct values decreased as PMI increased, while in the 4 °C lung, stable values were observed after 12 h.

Histology indicated that RNA could be extracted from tissues lacking nuclei after 8 days at 26 °C. Therefore, further bioanalysis was planned to confirm the RNA integrity number (RIN) in the future experiments. The RIN value is applied to electrophoretic RNA measurements, typically obtained using capillary gel electrophoresis, and is based on a combination of different features that contribute information about the RNA’s integrity to provide a more universal measure. RIN has been demonstrated to be robust and reproducible in studies comparing it to other RNA integrity calculation algorithms, cementing its position as a preferred method of determining the quality of RNA to be analyzed. In terms of delta Ct value, *Gapdh* exhibited the lowest values across all tissues and temperatures, suggesting potential differences in expression levels. However, the amplicon size was small for other housekeeping genes, which can affect the amplicon size to some extent. These findings provide preliminary evidence that RNA analysis may contribute to or has potential utility in estimating PMI, although further validation and model developments are required to confirm its applicability.

In addition, recent studies have highlighted that miRNAs exhibit higher stability than mRNAs within the first 24–48 h after death, suggesting their potential as more reliable post-mortem markers. For instance, several reports have demonstrated that specific miRNAs remain detectable and stable during early PMIs, making them promising candidates for normalization or estimation purposes [23]. Future studies could include a pilot miRNA cohort to compare their degradation kinetics with mRNAs and further refine molecular estimation models for PMIs.

Similarly, lncRNAs and circRNAs have also been reported to show remarkable stability in post-mortem tissues. Unlike mRNAs, these non-coding RNAs possess secondary structures or circular conformations that protect them from exonuclease degradation. Recent studies have demonstrated that circRNAs in particular remain stable under extended PMI conditions and could serve as complementary molecular indicators alongside mRNAs and miRNAs [24]. Incorporating lncRNA and circRNA profiling in future studies could therefore enhance the robustness of RNA-based PMI estimation.

This study has several limitations that should be acknowledged. Although the importance of the RIN was discussed, the actual RIN measurement was not performed in this experiment. In future studies, electrophoretic RNA integrity analysis will be included to validate RNA quality more precisely. In addition, primer efficiencies, standard curves, melt curve analyses, and DNA contamination controls were not presented in this study but will be addressed in future experiments to improve the quantitative reliability of RT-qPCR results. Despite these limitations, the findings provide valuable baseline information for understanding post-mortem RNA degradation under different temperature and PMI conditions. Although this study provides valuable baseline information on RNA degradation patterns in rat tissues under different PMI and temperature conditions, direct extrapolation of these findings to human forensic cases should be approached with caution. Species-specific differences in RNA stability and post-mortem biochemical responses may influence degradation dynamics. Therefore, further validation using human samples or simulated forensic conditions is required to confirm the applicability of these results to real-world forensic scenarios.

## 4. Materials and Methods

This study was approved by the Animal Experimental Ethics Committee of the Catholic University of Pusan (CUP AEC 2023-001). A total of 72 male Sprague Dawley (SD) rats aged six weeks, weighing 180–200 g, were randomly divided into 24 groups (three rats per group). According to animal experimental ethics, a CO_2_ chamber was used to suffocate the rats, which were placed in a sealed container for each group and kept at temperatures corresponding to the average summer, 26 °C and winter, 4 °C temperatures of the Republic of Korea. They were then taken out according to 12 PMI time points namely, 0 h, 6 h, 12 h, 24 h, 36 h, 48 h, 4 days, 6 days, 8 days, 10 days, 14 days, and 21 days to harvest organ tissues.

The organ samples were immersed in 10% neutral-buffered formalin for fixation for 24 h to preserve cellular and structural components. The fixed organ samples were sectioned to the desired size and processed to pass through paraffin wax. Paraffin wax stiffens the organ sample and serves as the primary material for creating paraffin blocks. Organ sections processed with paraffin wax were used to create blocks that were used for organ slicing. Thin organ sections were made from the paraffin blocks were attached to the slides to be stained. Then, the organ slides were exposed to hematoxylin and eosin (H&E) staining agents. Hematoxylin stains the cell nuclei blue, while Eosin stains cellular structures and cytoplasm pink. This staining process allows for the visual differentiation of cellular and structural elements in the organ. The stained organ slides were examined at ×200 using a Leica S9D Stereo Microscopes (Leica Microsystems, Wetzlar, Germany).

The total RNA isolation from stabilized rat organ tissue samples were performed using RiboEx^TM^ (GeneAll, Seoul, Republic of Korea). First, the tissue samples were weighed and cut to 0.01 g for uniform size. The samples were homogenized with 1 mL RiboEx^TM^ lysis solution (GeneAll) and the homogenate was incubated for 5 min at room temperature. Next, 0.2 mL of chloroform was added, and the sample was shaken vigorously for 15 s before, being stored for 2 min at room temperature. The sample was centrifuged at 12,000× *g* for 15 min at 4 °C, then the aqueous phase was transferred to a fresh tube. After centrifugation, 0.5 mL of isopropyl alcohol was added and the samples were incubated for 10 min at room temperature. They were then centrifuged at 12,000× *g* for 10 min at 4 °C, and the supernatant was discarded. To wash the RNA pellet, 1 mL of 75% ethanol was added and it was centrifuged at 7500× *g* for 5 min. The supernatant and ethanol were carefully discarded, and the RNA pellet was air-dried for 5 min. The sample dissolve RNA in DEPC-treated water by incubation for 10 min at 56 °C. Finally, the concentration and purity of the isolated total RNA were measured using a NanoDrop 2000 spectrophotometer (Thermo Fisher Scientific, Waltham, MA, USA). The A260/280 ratio of all RNA samples ranged from 1.8 to 2.2, indicating acceptable purity for downstream analyses. The samples were stored at −80 °C for later use.

Four genes—5S rRNA, *B2m*, *Gapdh*, and *Sort1*—were selected as target markers for post-mortem RNA analysis. 5S rRNA was chosen as a highly stable structural RNA, while *B2m* and *Gapdh* are well-known housekeeping genes used for normalization across tissues. *Sort1* was included to represent a functionally tissue-specific gene. Although it is mainly expressed in the brain, its detectable expression in peripheral organs enabled comparison of degradation dynamics between tissue-specific and housekeeping genes.

cDNA was synthesized using a Moloney Murine Leukemia Virus (M-MLV) Reverse Transcriptase kit (Invitrogen, Carlsbad, CA, USA) according to the manufacturer’s instructions. Before the experiment, total RNA was diluted to 100 ng/uL. First, 10 uL of total RNA was added to a master mix containing 1 uL of 10 mM dNTP mix (10 mM each of dATP, dGTP, dCTP, and dTTP at neutral pH), 1 uL of DEPC, and 1 uL of random primer in a PCR tube, and the reaction mixture was incubated at 65 °C for 5 min before being quickly chilled on ice and spun down. Next, a mixture of 4 uL of 5× First-Strand Synthesis Buffer, 2 uL of 100 mM dithiothreitol (DTT), and 1 uL of M-MLV RT was added to the previous reaction mixture in PCR tubes and then incubated at 25 °C for 10 min, 37 °C for 50 min, and 70 °C for 15 min. All reactions were performed using a SimpliAmp (Life Technologies, Carlsbad, CA, USA) thermal cycler. The synthesized final cDNA samples were stored at −20 °C for later use.

For each reaction in the PCR plates, 10 uL of SYBR Green Real-Time PCR Master Mix (TOYOBO, Osaka, Japan), 1 uL each of 10 pmol sense and anti-sense oligonucleotide primer set, 5 uL of ultra-pure DNase-/RNase-free distilled water, and 3 uL of synthesized cDNA were used for a final volume of 20 uL. The RT-qPCR primer pairs were used to detect 5S rRNA and three types of housekeeping gene mRNAs, namely, *B2m*, *Gapdh*, and *Sort1* (Table 1). The thermal cycling conditions were 60 s at 95 °C, followed by 35 cycles of 10 s at 95 °C and 30 s at 60 °C. The relative expression of each target marker looked at the quality of RNA by determining the Ct, that is, the number of PCR cycles required for the fluorescence to be significantly higher than the background. Melting curve analysis was performed at the end of each run to verify the specificity of amplification. No-template controls (NTC) and no-RT controls were included in each assay using nuclease-free distilled water instead of cDNA to confirm the absence of contamination and non-specific amplification.

Statistical analysis was performed using GraphPad Prism v5.00 (GraphPad Software, San Diego, CA, USA). One-way ANOVA followed by Tukey’s post hoc test was used to evaluate significant differences in Ct and delta Ct values among different PMIs and tissues. A *p*-value < 0.05 was considered statistically significant. Data are presented as mean ± standard deviation (SD) and were calculated for each group. Significant differences among tissues, temperatures, PMI, and housekeeping genes were analyzed based on Ct values obtained from RT-qPCR. Delta Ct values were calculated by normalizing *B2m*, *Gapdh*, and *Sort1* expression to 5S rRNA. Group comparisons were conducted using column-based replication and sub-column analysis across PMI time points.

## Figures and Tables

**Figure 1 ijms-26-11227-f001:**
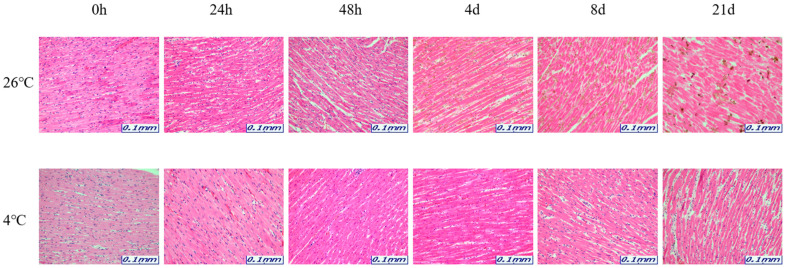
Changes in histological morphology in heart tissue according to PMI using light microscopy after H&E staining (×200).

**Figure 2 ijms-26-11227-f002:**
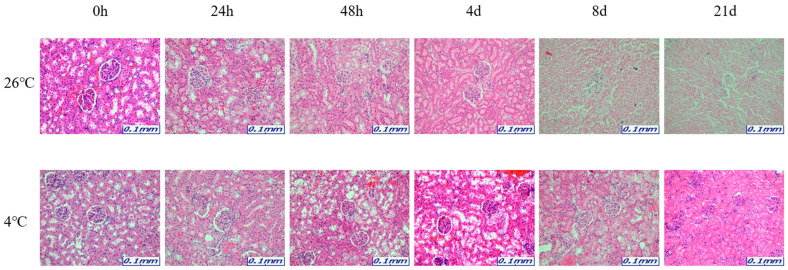
Changes in histological morphology in kidney tissue according to PMI using light microscopy after H&E staining (×200).

**Figure 3 ijms-26-11227-f003:**
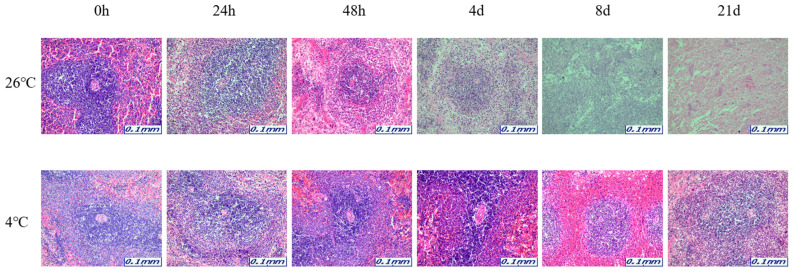
Changes in histological morphology in liver tissue according to PMI using light microscopy after H&E staining (×200).

**Figure 4 ijms-26-11227-f004:**
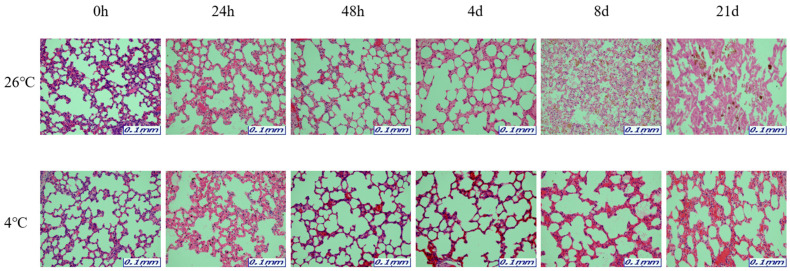
Changes in histological morphology in lung tissues according to PMI using light microscopy after H&E staining (×200).

**Figure 5 ijms-26-11227-f005:**
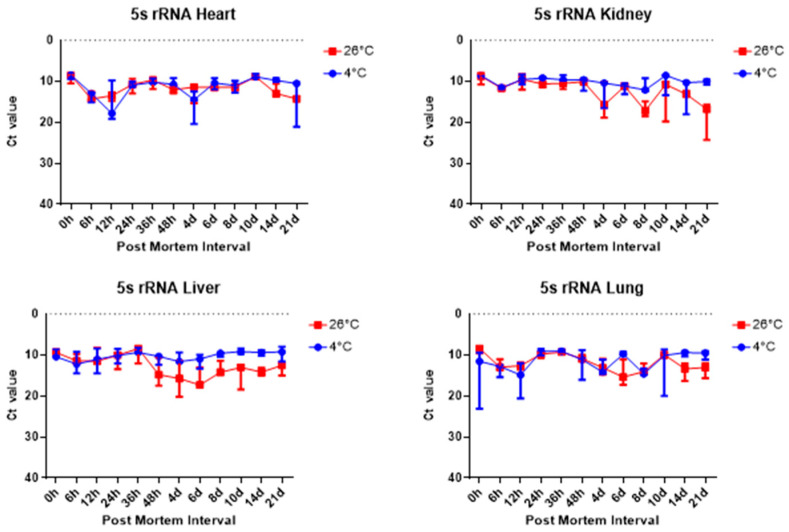
Changes in Ct values of 5S rRNA in heart, kidney, liver, and lung tissues according to PMI. Data are presented as median with range from three independent biological replicates.

**Figure 6 ijms-26-11227-f006:**
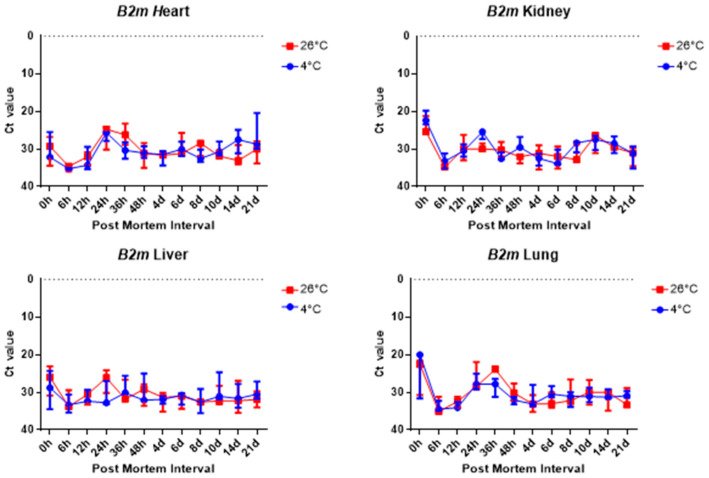
Changes in Ct values of *B2m* in heart, kidney, liver, and lung tissues according to PMI. Data are presented as median with range from three independent biological replicates.

**Figure 7 ijms-26-11227-f007:**
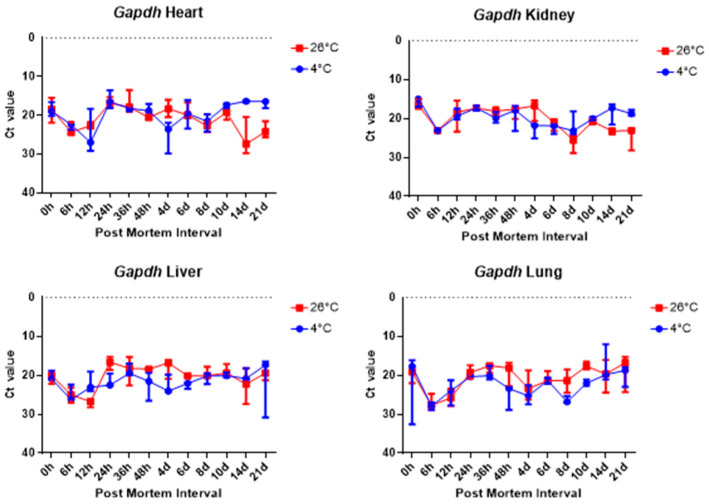
Changes in Ct values of *Gapdh* in heart, kidney, liver, and lung tissues according to PMI. Data are presented as median with range from three independent biological replicates.

**Figure 8 ijms-26-11227-f008:**
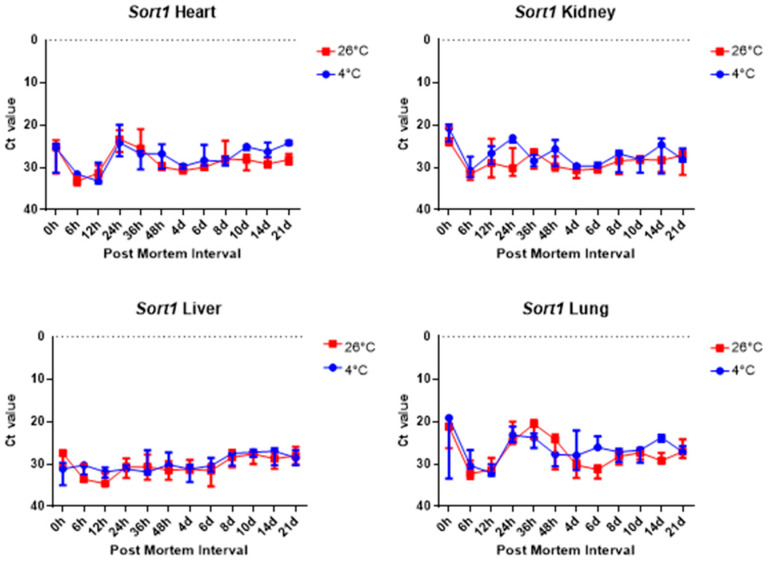
Changes in Ct values of *Sort1* in heart, kidney, liver, and lung tissues according to PMI. Data are presented as median with range from three independent biological replicates.

**Figure 9 ijms-26-11227-f009:**
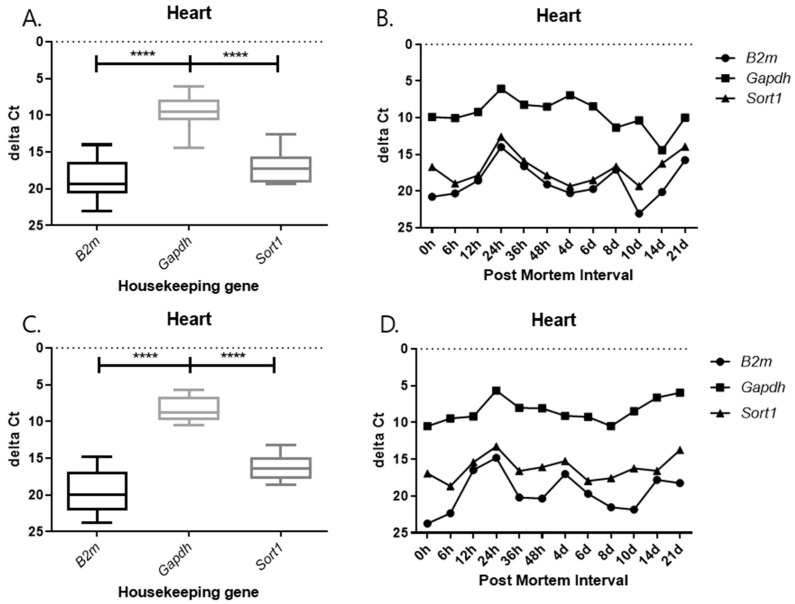
Changes in the delta Ct values of *B2m*, *Gapdh*, and *Sort1* genes in heart tissue according to PMI. (**A**,**B**) 26 °C; (**C**,**D**) 4 °C. **** *p*-value < 0.0001.

**Figure 10 ijms-26-11227-f010:**
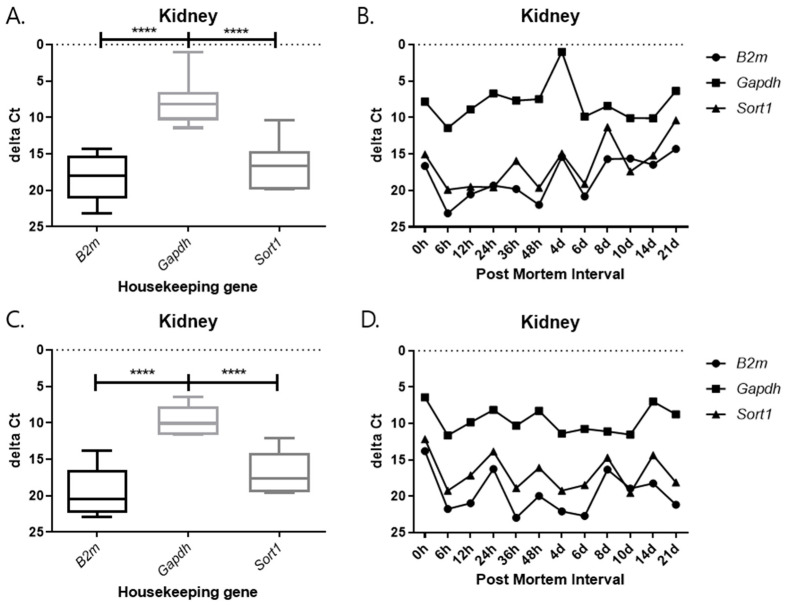
Changes in the delta Ct values of *B2m*, *Gapdh*, and *Sort1* genes in kidney tissue according to PMI. (**A**,**B**) 26 °C; (**C**,**D**) 4 °C. **** *p*-value < 0.0001.

**Figure 11 ijms-26-11227-f011:**
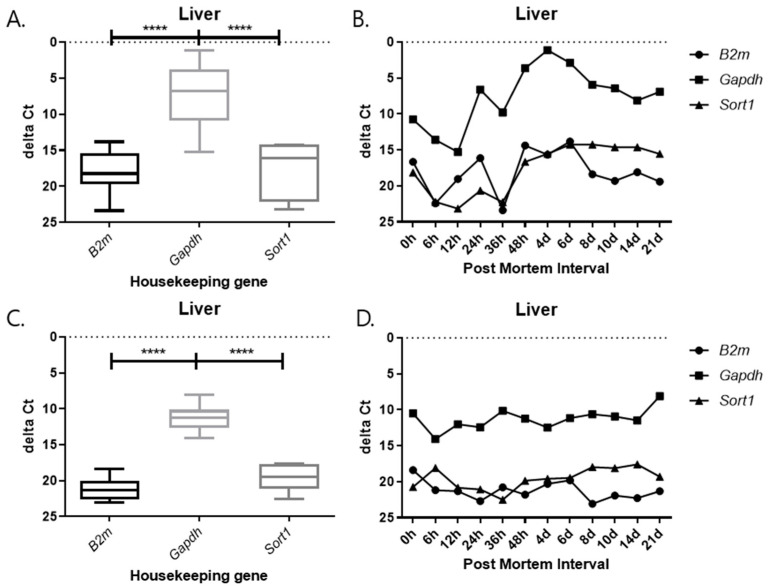
Changes in the delta Ct values of *B2m*, *Gapdh*, and *Sort1* genes in liver tissue according to PMI. (**A**,**B**) 26 °C; (**C**,**D**) 4 °C. **** *p*-value < 0.0001.

**Figure 12 ijms-26-11227-f012:**
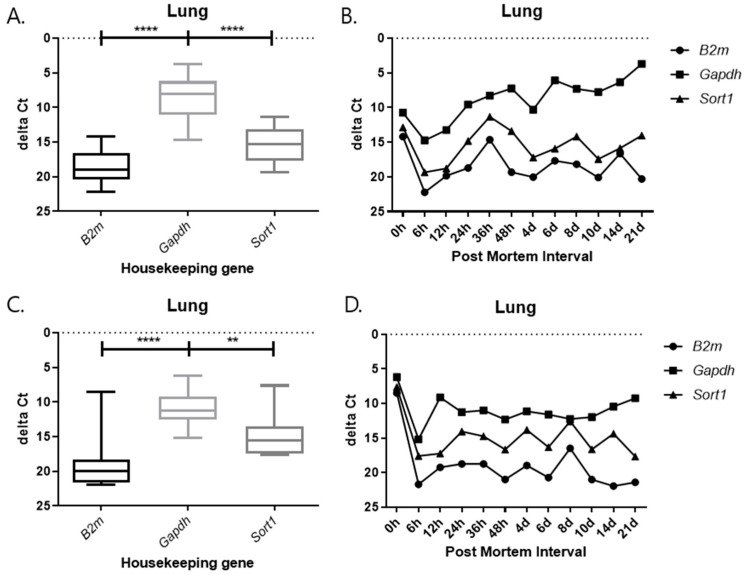
Changes in the delta Ct values of *B2m*, *Gapdh*, and *Sort1* genes in lung tissue according to PMI. (**A**,**B**) 26 °C; (**C**,**D**) 4 °C. **** *p*-value < 0.0001, ** *p*-value < 0.01.

**Table 1 ijms-26-11227-t001:** Oligo-nucleotide primer pairs for the RT-qPCR SYBR Green assay used in the present study.

Group	Target Genes	Gene Name	Primer Sequences (5′-3′)	Amplicon Size (bp)	Gene Accession No.	References
Internal reference gene	5S rRNA	5s Ribosomal RNA	F:ATCTCGTCTGATCTCGGAA	59	K01594	[15]
			R:TCTCCCATCCAAGTACTAACC			
Housekeeping gene	*B2m*	Beta-2 microglobulin	F:AGTAGGAGGTGCTCGATGAAG	148	NM_012512	
			R:TCCTGTAGAGCCAGCAACAGG			
	*Gapdh*	Glyceraldehyde-3-phosphate dehydrogenase	F:TGACAACTTTGGCATCGTGG	78	NM_017008	
			R:GGGCCATCCACAGTCTTCTG			
	*Sort1*	Sortilin 1	F:CTGACCAACAATACGCACCA	250	XM_134443	[16]
			R:AGTTCTCGGGACCAATAGCC			

## Data Availability

The original contributions presented in this study are included in the article. Further inquiries can be directed to the corresponding authors.

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
