# Peer review of "Evaluation of RNA Stability and Molecular Biomarkers for Post-Mortem Interval Estimation in Rat Organs"

_ijms, 2025, doi:10.3390/ijms262211227_

Round 1
Reviewer 1 Report
Comments and Suggestions for Authors
This study addresses a timely and forensically relevant question: the utility of RNA-based biomarkers for estimating the post-mortem interval (PMI) under controlled environmental conditions. The experimental design is generally sound, employing multiple time points, two temperature regimes (26°C and 4°C), and four major organs from Sprague-Dawley rats. The inclusion of both histological assessment and RT-qPCR–based molecular analysis strengthens the multidisciplinary approach. However, several methodological, analytical, and interpretive issues must be addressed before the manuscript can be considered for publication.
Major Comments
- While the authors mention RIN in the Discussion (lines 262–269), no actual RIN values or gel/electropherogram data are presented in the Results or Methods. Given that RNA degradation is central to the study’s premise, omitting objective RNA integrity metrics (e.g., via Bioanalyzer or TapeStation) significantly weakens the conclusions. The NanoDrop data alone cannot distinguish between intact RNA and fragmented nucleic acids.so its recommended Either include RIN or equivalent integrity data, or explicitly acknowledge this limitation and temper conclusions accordingly.
- The use of 5S rRNA as a reference gene is unconventional in RT-qPCR normalization, especially for mRNA targets (B2m, Gapdh, Sort1). While 5S rRNA is structurally stable, it is transcribed by RNA Pol III and not co-regulated with mRNA (Pol II transcripts), violating a key assumption of reference gene validity. The authors do not provide evidence that 5S rRNA expression remains constant across PMI and temperature conditions, only that its Ct values are low. So its better to validate 5S rRNA stability using algorithms like geNorm, NormFinder, or BestKeeper. Alternatively, justify its use with literature specific to post-mortem RNA decay or consider using multiple validated reference genes.
- The manuscript states that “Ct values generally increased with prolonged PMI” (line 111), yet Figure 5C shows a decrease in Ct for liver 5S rRNA at 26°C after 48 hours, interpreted as increased stability. A decreasing Ct implies higher apparent abundance, which contradicts expected RNA degradation unless there is selective preservation or technical artifact (e.g., inhibition loss over time). This requires clarification. Similarly, the claim that “Gapdh exhibited the lowest ΔCt values” (line 270) is ambiguous low ΔCt could reflect either high target expression or poor reference gene performance.
- The Methods state that statistical analysis was performed using GraphPad Prism but do not specify which tests were used (e.g., ANOVA, repeated measures, post-hoc corrections). With 12 time points, two temperatures, four organs, and multiple genes, appropriate multivariate or mixed-effects models are needed. The current description (“column-based replication”) is insufficient. So, provide full statistical methodology, including tests, p-value adjustments, and effect sizes.
Minor Comments
- Figures 5–12 are referenced but not included in the provided PDF. Review cannot assess data visualization quality. Ensure all figures are high-resolution and include error bars, n-values, and statistical annotations in the final version.
- While primer sequences are provided, there is no mention of melt curve analysis or amplicon verification (e.g., sequencing). Given the degraded nature of post-mortem RNA, non-specific amplification is a concern.
- The rationale for including Sort1 described as “brain-specific” in heart, kidney, liver, and lung is unclear. If it is not expressed in these tissues, its utility as a biomarker is questionable.
- The temperatures (4°C and 26°C) are justified as approximating Korean winter/summer, but real-world forensic cases involve fluctuating temperatures, humidity, and microbial exposure. A brief discussion of translational limitations would strengthen the Discussion.
- Typographical and Formatting Issues: Several grammatical errors and inconsistent formatting (e.g., “PMI” vs. “post-mortem interval”, spacing around parentheses) should be corrected.
The study has merit and addresses an important gap in forensic RNA biomarker research. However, the concerns regarding RNA integrity validation, reference gene appropriateness, statistical robustness, and data interpretation must be addressed before the work can be accepted.
Author Response
Comment 1 : While the authors mention RIN in the Discussion (lines 262–269), no actual RIN values or gel/electropherogram data are presented in the Results or Methods. Given that RNA degradation is central to the study’s premise, omitting objective RNA integrity metrics (e.g., via Bioanalyzer or TapeStation) significantly weakens the conclusions. The NanoDrop data alone cannot distinguish between intact RNA and fragmented nucleic acids.so its recommended Either include RIN or equivalent integrity data, or explicitly acknowledge this limitation and temper conclusions accordingly.
Response 1 : Thank you for pointing this out. I agree with this comment. Therefore, I have page 11, line 273, page 12, line 301-305
Comment 2 : The use of 5S rRNA as a reference gene is unconventional in RT-qPCR normalization, especially for mRNA targets (B2m, Gapdh, Sort1). While 5S rRNA is structurally stable, it is transcribed by RNA Pol III and not co-regulated with mRNA (Pol II transcripts), violating a key assumption of reference gene validity. The authors do not provide evidence that 5S rRNA expression remains constant across PMI and temperature conditions, only that its Ct values are low. So its better to validate 5S rRNA stability using algorithms like geNorm, NormFinder, or BestKeeper. Alternatively, justify its use with literature specific to post-mortem RNA decay or consider using multiple validated reference genes.
Response 2 : Thank you for pointing this out. I agree with this comment. Therefore, I have page 11, line 254-265
Comment 3 : The manuscript states that “Ct values generally increased with prolonged PMI” (line 111), yet Figure 5C shows a decrease in Ct for liver 5S rRNA at 26°C after 48 hours, interpreted as increased stability. A decreasing Ct implies higher apparent abundance, which contradicts expected RNA degradation unless there is selective preservation or technical artifact (e.g., inhibition loss over time). This requires clarification. Similarly, the claim that “Gapdh exhibited the lowest ΔCt values” (line 270) is ambiguous low ΔCt could reflect either high target expression or poor reference gene performance.
Response 3 : Thank you for pointing this out. I agree with this comment. Therefore, I have page 11, line 261-265
Comment 4 : The Methods state that statistical analysis was performed using GraphPad Prism but do not specify which tests were used (e.g., ANOVA, repeated measures, post-hoc corrections). With 12 time points, two temperatures, four organs, and multiple genes, appropriate multivariate or mixed-effects models are needed. The current description (“column-based replication”) is insufficient. So, provide full statistical methodology, including tests, p-value adjustments, and effect sizes.
Response 4 : Thank you for pointing this out. I agree with this comment. Therefore, I have page 13, line 390-394

Reviewer 2 Report
Comments and Suggestions for Authors
Dear Authors, I read your article with interest, but it requires extensive revision before it can be considered for publication. Here are some comments that I hope will help resolve some ambiguities:
- In the abstract, some sentences are unclear ("qualification of RNA," "the main purpose of this study"). You should improve your style and remove repetitions and vagueness.
- The authors state that nuclei remain "intact at 4°C for up to 21 days": I suggest supporting this statement with a quantitative cytomorphic scoring. The literature shows that antigenic and nuclear preservation depend on tissue, fixation, and time, so I suggest introducing a score or other support.
- In the introduction, the authors state that "cell proliferation and tumor formation" may be consequences of increasing PMI, but this is absolutely incorrect. I suggest reviewing this information and replacing it with correct information.
- The word "Henssage" is spelled Henssge.
- The introduction does not include the most recent works in the literature on miRNAs and circRNAs as PMI markers. I ask you to supplement the text with this information (references to be added in the text: 10.3390/ijms25179207, 10.1007/s00414-025-03590-3).
- The manuscript is missing essential elements such as RIN, A260/280 purity with ranges, primer efficiencies, standard curve, melt curves, DNA contamination control, etc.). The authors should integrate this information. If such information is not available, at least a "Limitations" section should be created to clearly state this.
- Regarding the choice of genes, Sort1 is described as "tissue-specific (brain)," but the analysis is performed on heart/kidney/liver/lung. Therefore, I ask the authors to justify their choice.
- Regarding RT-qPCR, as mentioned, the melting curves and no-RT/no-template controls are missing.
- The definition of ΔCt provided by the authors in the text also seems imprecise and repetitive (lines 198–208). I suggest the authors clearly define ΔCt = Ct(target) − Ct(reference).
- The authors describe "visual" differences in Ct without correct confidence intervals or p-values.
- Figures 5–8 do not have complete legends (n, error bars = SD or SEM, units) and do not report all conditions. I suggest the authors revise them for clarity.
- Regarding the authors' claims about purity, "similar in all organs because of phenol," I note that this inference is not always correct, as purity also depends on the tissue and carryover.
- The discussion fails to consider that miRNAs remain more stable in the first 24–48 hours and may be suitable as references/estimators; include a critical comparison and, if possible, a pilot miRNA cohort. Furthermore, the authors could briefly compare the evidence from other types of RNA, such as mRNAs (literature review: 10.3390/ijms25158185).
- The same consideration for lncRNAs and circRNAs (10.1038/s41598-024-70678-y, 10.1038/s41598-025-07998-0).
- I suggest the authors to provide raw data for reproducibility, particularly raw Ct values, efficiencies, and standard curves.
- Authors should specify that transferring data from animal to human samples, especially in forensic settings, requires species-specific validation in real-world scenarios. I suggest authors include this information in the limitations section or in the discussion.
- In their conclusions, the authors argue that RNA analysis could be useful for determining PMI, but they do not present a model, so I suggest limiting the strength of this assertion.
- In the Methods section, regarding euthanasia, the sex and weight of the rats, fixation time, etc, are not specified.
English needs extensive revision
Author Response
Comment 1 : In the abstract, some sentences are unclear ("qualification of RNA," "the main purpose of this study"). You should improve your style and remove repetitions and vagueness.
Response 1 : Thank you for pointing this out. I agree with this comment. Therefore, I have page 1, line 24-40
Comment 2 : The authors state that nuclei remain "intact at 4°C for up to 21 days": I suggest supporting this statement with a quantitative cytomorphic scoring. The literature shows that antigenic and nuclear preservation depend on tissue, fixation, and time, so I suggest introducing a score or other support.
Response 2 : Thank you for pointing this out. These comments are written by looking at a picture of the organization slide.
Comment 3 : In the introduction, the authors state that "cell proliferation and tumor formation" may be consequences of increasing PMI, but this is absolutely incorrect. I suggest reviewing this information and replacing it with correct information.
Response 3 : Thank you for pointing this out. I agree with this comment. Therefore, I have page 2, line 77-81
Comment 4 : The word "Henssage" is spelled Henssge.
Response 4 : Thank you for pointing this out. I agree with this comment. Therefore, I have page 2, line 53, 56
Comment 5 : The introduction does not include the most recent works in the literature on miRNAs and circRNAs as PMI markers. I ask you to supplement the text with this information (references to be added in the text: 10.3390/ijms25179207, 10.1007/s00414-025-03590-3).
Response 5 : Thank you for pointing this out. I agree with this comment. Therefore, I have page 3, line 111-115
Comment 6 : The manuscript is missing essential elements such as RIN, A260/280 purity with ranges, primer efficiencies, standard curve, melt curves, DNA contamination control, etc.). The authors should integrate this information. If such information is not available, at least a "Limitations" section should be created to clearly state this.
Response 6 : Thank you for pointing this out. I agree with this comment. Therefore, I have page 11, line 291-307, page 12, line 343-344
Comment 7 : Regarding the choice of genes, Sort1 is described as "tissue-specific (brain)," but the analysis is performed on heart/kidney/liver/lung. Therefore, I ask the authors to justify their choice.
Response 7 : Thank you for pointing this out. I agree with this comment. Therefore, I have page 12, line 349-351
Comment 8 : Regarding RT-qPCR, as mentioned, the melting curves and no-RT/no-template controls are missing.
Response 8 : Thank you for pointing this out. I agree with this comment. Therefore, I have page 13, line 372-376
Comment 9 : The definition of ΔCt provided by the authors in the text also seems imprecise and repetitive (lines 198–208). I suggest the authors clearly define ΔCt = Ct(target) − Ct(reference).
Response 9 : Thank you for pointing this out. I agree with this comment. Therefore, I have page 7, line 189-196
Comment 10 : The authors describe "visual" differences in Ct without correct confidence intervals or p-values.
Response 10 : Thank you for pointing this out. I agree with this comment. Therefore, I have page 13, line 377-381
Comment 11 : Figures 5–8 do not have complete legends (n, error bars = SD or SEM, units) and do not report all conditions. I suggest the authors revise them for clarity.
Response 11 : Thank you for pointing this out. I agree with this comment. Therefore, I have page 6-7
Comment 12 : Regarding the authors' claims about purity, "similar in all organs because of phenol," I note that this inference is not always correct, as purity also depends on the tissue and carryover.
Response 12 : Thank you for pointing this out. But I can't find the contents, so if you tell me the line, I'll revise it
Comment 13 : The discussion fails to consider that miRNAs remain more stable in the first 24–48 hours and may be suitable as references/estimators; include a critical comparison and, if possible, a pilot miRNA cohort. Furthermore, the authors could briefly compare the evidence from other types of RNA, such as mRNAs (literature review: 10.3390/ijms25158185).
Response 13 : Thank you for pointing this out. I agree with this comment. Therefore, I have page 11, line 279-282
Comment 14 : The same consideration for lncRNAs and circRNAs (10.1038/s41598-024-70678-y, 10.1038/s41598-025-07998-0).
Response 14 : Thank you for pointing this out. I agree with this comment. Therefore, I have page 11, line 283-290
Comment 15 : I suggest the authors to provide raw data for reproducibility, particularly raw Ct values, efficiencies, and standard curves.
Response 15 : Thank you for pointing this out. I agree with this comment. Therefore, I have page 11, line 291-307
Comment 16 : Authors should specify that transferring data from animal to human samples, especially in forensic settings, requires species-specific validation in real-world scenarios. I suggest authors include this information in the limitations section or in the discussion.
Response 16 : Thank you for pointing this out. I agree with this comment. Therefore, I have page 11, line 291-307
Comment 17 : In their conclusions, the authors argue that RNA analysis could be useful for determining PMI, but they do not present a model, so I suggest limiting the strength of this assertion.
Response 17 : Thank you for pointing this out. I agree with this comment. Therefore, I have page 11, line 291-307
Comment 18 : In the Methods section, regarding euthanasia, the sex and weight of the rats, fixation time, etc, are not specified.
Response 18 : Thank you for pointing this out. I agree with this comment. Therefore, I have page 12, line 310-313

Round 2
Reviewer 1 Report
Comments and Suggestions for Authors
The revised manuscript shows clear improvement. The authors have adequately addressed all of my previous comments. I have no additional concerns, and the manuscript is ready for publication.
Comments on the Quality of English LanguageNo
Reviewer 2 Report
Comments and Suggestions for Authors
I thank the authors for making the requested changes. The text still requires improvement in English.
Comments on the Quality of English LanguageEnglish needs further revision